# MAD1 upregulation sensitizes to inflammation-mediated tumor formation

**Sarah E. Copeland**[1], **Santina M. Snow**[2,3,4], **Jun Wan**[5], **Kristina A. Matkowskyj**[6,7], **Richard B. Halberg**[3,4,7], **Beth A. Weaver**[3,7,8] *

1 Molecular and Cellular Pharmacology Graduate Training Program, University of Wisconsin-Madison, Madison, Wisconsin, United States of America, 2 Cancer Biology Graduate Training Program, University of Wisconsin-Madison, Madison, Wisconsin, United States of America, 3 Department of Oncology/McArdle Laboratory for Cancer Research, University of Wisconsin–Madison, Madison, Wisconsin, United States of America, 4 Department of Medicine, University of Wisconsin-Madison, Madison, Wisconsin, United States of America, 5 Physiology Graduate Training Program, University of Wisconsin-Madison, Madison, Wisconsin, United States of America, 6 Department of Pathology and Laboratory Medicine, University of Wisconsin-Madison, Madison, Wisconsin, United States of America, 7 Carbone Cancer Center, University of Wisconsin-Madison, Madison, Wisconsin, United States of America, 8 Department of Cell and Regenerative Biology, University of Wisconsin-Madison, Madison, Wisconsin, United States of America

* baweaver@wisc.edu

**Data Availability Statement:** The authors confirm that all data underlying the findings are fully available without restriction. All relevant data are

## Abstract

Mitotic Arrest Deficient 1 (gene name *MAD1L1*), an essential component of the mitotic spindle assembly checkpoint, is frequently overexpressed in colon cancer, which correlates with poor disease-free survival. MAD1 upregulation induces two phenotypes associated with tumor promotion in tissue culture cells–low rates of chromosomal instability (CIN) and destabilization of the tumor suppressor p53. Using CRISPR/Cas9 gene editing, we generated a novel mouse model by inserting a doxycycline (dox)-inducible promoter and HA tag into the endogenous mouse *Mad1l1* gene, enabling inducible expression of HA-MAD1 following exposure to dox in the presence of the reverse tet transactivator (rtTA). A modest 2-fold overexpression of MAD1 in murine colon resulted in decreased p53 expression and increased mitotic defects consistent with CIN. After exposure to the colon-specific inflammatory agent dextran sulfate sodium (DSS), 31% of mice developed colon lesions, including a mucinous adenocarcinoma, while none formed in control animals. Lesion incidence was particularly high in male mice, 57% of which developed at least one hyperplastic polyp, adenoma or adenocarcinoma in the colon. Notably, mice expressing HA-MAD1 also developed lesions in tissues in which DSS is not expected to induce inflammation. These findings demonstrate that MAD1 upregulation is sufficient to promote colon tumorigenesis in the context of inflammation in immune-competent mice.

## Author summary

Worldwide, colorectal cancer is the third most commonly diagnosed cancer and the second-highest cause of cancer-related deaths. A better understanding of the molecular causes of colorectal cancer could provide novel therapeutic targets for this disease. Mitotic

within the paper and its Supporting Information files.

**Funding:**
This work was supported, in part, by National Institutes of Health (grant R01CA270133 to BAW), the University of Wisconsin UW2020 award for Advancing CRISPR-mediated Genome Editing Technology at UW-Madison to Model Human Disease, and Wisconsin Dual Sports Riders (to SMS and RBH). The funders had no role in study design, data collection and analysis, decision to publish, or preparation of the manuscript.

**Competing interests:** The authors have declared that no competing interests exist.

Arrest Deficient-1 (MAD) is commonly overexpressed in colorectal cancers, and elevated expression of MAD1 is associated with worse prognosis. Here we generated a new mutant mouse with inducible, modest overexpression of MAD1. First identified for its role in mitosis, MAD1 is required to ensure proper chromosome segregation. Elevated expression of MAD1 causes chromosome mis-segregation in the colons of these mice. During interphase, overexpressed MAD1 destabilizes the well-known tumor suppressor protein, p53. Consistent with this, mouse colons modestly overexpressing MAD1 show decreased expression of p53. Since inflammation is a risk factor for colorectal cancer, we induced inflammation in the colon using dextran sulfate sodium (DSS) and found that MAD1 overexpression significantly increased the incidence of colon tumors. Like inflammation-mediated colon tumors in humans, in the mice these tumors were more common in males than females. This work shows that MAD1 overexpression can promote colon cancer and suggests that MAD1 may be a novel drug target for this disease.

## Introduction

Colorectal cancer is the third leading cause of cancer-related death in the US [1], underscoring the necessity of identifying and validating drivers of the disease that may serve as novel therapeutic targets. Overexpression of mitotic genes, including *MAD1L1*, is commonly observed in colorectal and other cancers [2]. Many mitotic genes are cell cycle regulated, so their apparent upregulation may simply be due to the increased proliferation of cancerous versus normal tissue. However, MAD1 is not cell cycle regulated at the mRNA or protein level [3,4], suggesting that MAD1 overexpression confers a selective advantage to tumors. Consistent with this, MAD1 overexpression causes at least two phenotypes associated with tumor-promoting effects. First, MAD1 overexpression induces low rates of recurrent chromosome missegregation termed chromosomal instability (CIN) [3]. Low CIN promotes tumorigenesis in some contexts [5,6], though it has also been reported to activate the immune system, which limits tumorigenesis by eliminating aneuploid cells [7,8]. Second, MAD1 overexpression results in destabilization of the tumor suppressor p53 [9].

The best characterized function of MAD1 is in preventing CIN, a feature present in ~50% of colon cancers [5]. *MAD1* was one of the original mitotic spindle assembly checkpoint genes discovered in budding yeast [10]. It plays a well-established, evolutionarily conserved role in the mitotic checkpoint [11–14], which ensures that cells segregate DNA equally during mitosis to produce two genetically identical daughter cells [15,16]. Chromosomes, which enter mitosis as connected pairs of replicated sister chromatids, are sorted on a mitotic spindle composed of microtubules. The mitotic checkpoint delays separation of the sister chromatids until each sister chromatid pair has made proper attachments to the mitotic spindle. MAD1 functions in the mitotic checkpoint by recruiting its binding partner MAD2 from a large inactive soluble pool to the microtubule attachment sites of unattached chromatids (kinetochores) [17,18], where MAD2 undergoes a conformational change that allows it to inhibit the Anaphase Promoting Complex/Cyclosome (APC/C), an E3 ubiquitin ligase whose activity drives mitotic progression [15,16,19,20]. MAD1 overexpression weakens mitotic checkpoint signaling by binding MAD2 in the cytoplasm and reducing the number of free MAD2 molecules available for conversion into APC/C inhibitors at kinetochores [3,21]. Thus, increased MAD1 expression causes cells to enter anaphase before all sister chromatid pairs are properly attached to the mitotic spindle, resulting in CIN.

Though MAD1 was originally characterized for its role in mitosis, it is expressed throughout the cell cycle [3]. Overexpressed MAD1 localizes to PML (promyelocytic leukemia) nuclear bodies and destabilizes p53 [9]. p53 levels are normally low due to ubiquitination and degradation [22,23]. In response to cellular stress, PML sequesters the E3 ubiquitin ligase MDM2 into nucleoli, physically separating it from p53 and allowing p53 levels to accumulate [24]. Overexpressed MAD1 displaces MDM2 from PML, allowing MDM2 to continue ubiquitinating p53 [9].

Though MAD1 is commonly overexpressed in colorectal cancer and has potential tumor-promoting activities, it has remained unknown whether upregulation of MAD1 is sufficient to initiate tumorigenesis in immune-competent animals. Here, we describe the generation of a novel genetically engineered mouse model (GEMM) with doxycycline (dox)-inducible expression of endogenous *Mad1l1*. We show that modest MAD1 overexpression increased mitotic defects consistent with CIN, decreased p53, and sensitized immune-competent mice to colon tumors in the context of inflammation.

## Results

### *MAD1L1* overexpression is common in colon cancer where it serves as a marker of poor prognosis

When compared to normal samples, the gene encoding MAD1, *MAD1L1*, is overexpressed in 25% of colon adenocarcinomas in The Cancer Genome Atlas (TCGA) PanCancer Atlas (Fig 1A). Reduced expression, mutation and amplification of *MAD1L1* are rare (5.7%, 1.2% and 0.3%, respectively; Fig 1A) in colon cancer. Kaplan-Meier analysis of 1336 colon cancer patients (kmplot.com [25]) demonstrated that high expression of *MAD1L1* associates with significantly worse prognosis as measured by relapse free survival (Fig 1B). Similarly, post-progression survival was worse in patients with tumors expressing high levels of *MAD1L1* (Fig 1C) as compared to those with low levels. Thus, *MAD1L1* overexpression commonly occurs in colon cancer and correlates with poor patient outcome.

### Generation of dox-inducible HA-MAD1 mice

To interrogate the functional consequences of MAD1 upregulation *in vivo*, we used CRISPR/Cas9 to create a novel GEMM. We introduced a tetracycline (tet)/dox-inducible promoter into

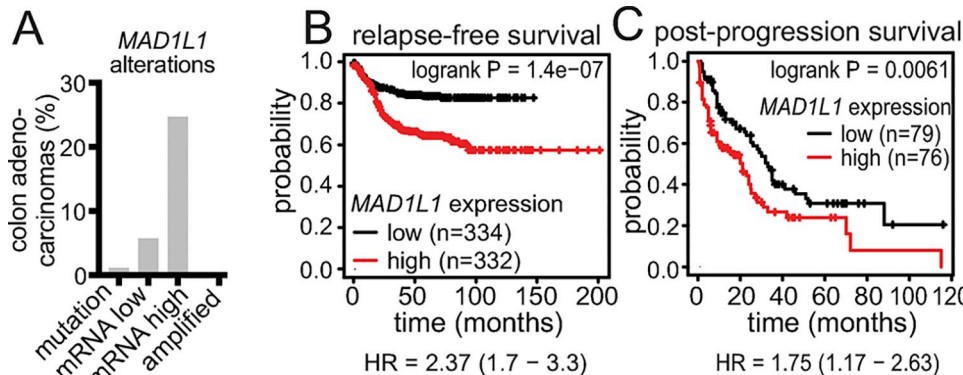

**Fig 1. Overexpression of *MAD1L1* is common in colon cancer, where it correlates with worse prognosis.** (A) Alterations in the gene encoding MAD1, *MAD1L1*, in 524 samples of normal versus colon adenocarcinoma from TCGA PanCancer Atlas (cancer study identifier: coadread_tcga_pan_can_atlas_2018). z threshold = 2.0. (B-C) Kaplan-Meier survival analysis (KMPlot [25]) showing high expression of *MAD1L1* mRNA is associated with worse relapse free survival in a cohort of 1336 patients (B) and post-progression survival in a cohort of 311 colon cancer patients (C). Q1 (lower quartile; "low") vs Q4 (upper quartile; "high") analysis is shown. HR = hazard ratio.

the endogenous *Mad1l1* locus, along with an N-terminal HA tag to facilitate detection of induced MAD1 (Figs 2A and A in S1 Text). Donor DNA used for homology directed repair, Cas9 protein, and two sgRNAs were injected into zygotes from C57BL/6J (B6) animals and transferred to a pseudopregnant female (Fig A in S1 Text). Next generation sequencing identified two founder mice with the precise edit and one with an additional 13 bp deletion in the intron following exon 2. Germline transmission from one mouse with the precise edit as well as the mouse with the 13 bp deletion occurred (Fig A in S1 Text). *Mad1l1* is essential [26], and we were unable to generate homozygous TetO-HA-MAD1 mice, consistent with dox-regulated control of the edited allele (Table A in S1 Text). Mice heterozygous for the TetO-HA-MAD1 edit were bred with B6 mice expressing rtTA3 from the CAG promoter, a strong synthetic promoter, inserted into the Rosa26 locus (JAX 029627 [27,28]) for expression in multiple tissues. Dox feed resulted in increased MAD1 mRNA but not protein in TetO-HA-MAD1$^{KI/+}$;Rosa26-CAG-rtTA3$^{KI/+}$ tissues (Fig B in S1 Text). We tested whether mice homozygous for the Rosa26-CAG-rtTA3 knock in allele expressed higher levels of rtTA3, resulting in increased MAD1 protein. However, mice homozygous for this allele, which are born at less than expected frequency and are poor breeders, did not show evidence for increased expression of HA-MAD1 (Fig B panels E-F in S1 Text). We therefore tested Rosa26-rtTA-M2 mice (JAX 006965 [29]), which express an optimized version of the reverse tet transactivator from the Rosa26 promoter, and can be maintained as homozygous lines. Two rounds of mating of TetO-HA-MAD1$^{KI/+}$ with Rosa26-rtTA-M2 mice produced TetO-HA-MAD1$^{KI/+}$;rtTA-M2$^{KI/KI}$ (hereafter HA-MAD1) animals (Fig 2B). HA-MAD1 expression was not detected in the absence of dox feed (Fig B panel H in S1 Text). In the presence of dox chow to induce expression in this Tet-On model (Fig 2C), these animals expressed HA-MAD1 protein in multiple tissues including spleen, colon, and skin (Fig 2D, upper band in MAD1 doublet), while control MAD1$^{+/+}$; rtTA-M2$^{KI/KI}$ (hereafter control) animals did not. Quantitation revealed that the amount of MAD1 upregulation at the protein level was modest (~2 fold; Fig 2E), though the increase in mRNA expression was higher (Fig 2F). These data demonstrate successful generation of a novel GEMM which modestly overexpresses MAD1 in response to dox.

## HA-MAD1 mice have increased mitotic defects and decreased p53

To determine whether 2-fold overexpression of MAD1 protein was sufficient to induce CIN and p53 degradation in colon, we supplied HA-MAD1 and control animals with dox feed *ad libitum* for 1 week and then harvested colon samples for immunofluorescence. The HA tag was readily apparent in the colonic epithelium of HA-MAD1, but not control mice, and showed the expected MAD1 localization pattern at the nucleus and nuclear envelope (Fig 3A). Total MAD1 levels were also increased and showed a similar localization (Fig 3B and 3C). We next tested whether p53 levels were decreased in the colon by HA-MAD1 expression. Immunoblotting with two different p53 antibodies revealed that subtle overexpression of MAD1 was sufficient to decrease p53 levels in the colon (Fig 3D and 3E). Quantitative immunofluorescence revealed that p53 expression was reduced in the colonic epithelium (Fig 3F and 3G).

We also tested whether ~2-fold overexpression of MAD1 protein was sufficient to reduce mitotic fidelity and induce mitotic defects consistent with a low rate of CIN in colon. Examination of H&E-stained colon sections revealed a substantial increase in abnormal mitotic figures. As expected in MAD1 overexpressing cells with a weakened mitotic checkpoint, HA-MAD1 mice had a significantly higher incidence of lagging chromosomes (that lag behind the segregating masses of DNA in anaphase and telophase) than control mice (Fig 3H and 3I). These data demonstrate that a modest increase in MAD1 expression in the colonic epithelium is sufficient to decrease p53 expression and increase mitotic defects that are consistent with CIN.

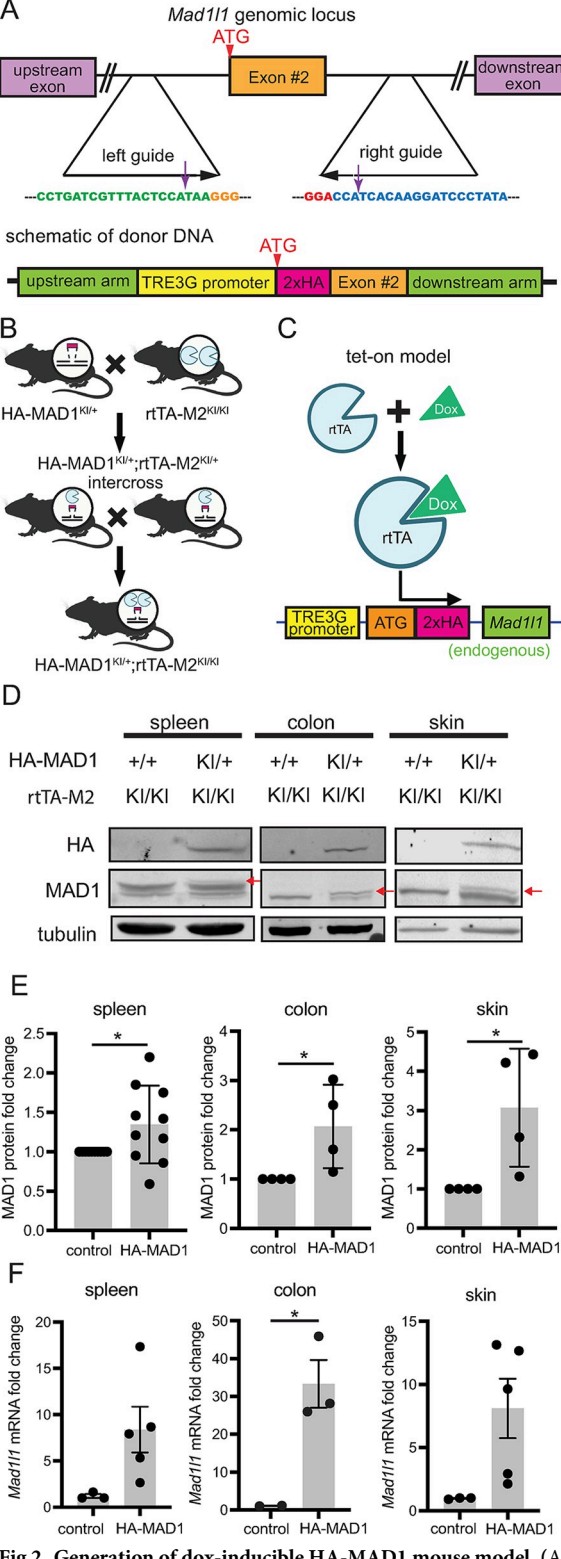

**Fig 2. Generation of dox-inducible HA-MAD1 mouse model.** (A) Schematic showing strategy for editing the endogenous mouse *Mad1l1* locus (top) and the donor DNA template used for homology directed repair (bottom). (B) Breeding scheme used to obtain mice with both the inducible *Mad1l1* allele and the rtTA-M2 reverse tetracycline transactivator protein. KI = Knock In. (C) Schematic shows dox-inducible Tet-ON expression of endogenous mouse *Mad1l1*. (D) Immunoblots show inducible expression of HA-MAD1 after 1 week of 6 mg/kg dox feed. Red arrows

indicate HA-MAD1. (E) Quantification of MAD1 protein levels (mean +/- SE). Each dot represents tissue from a different animal. (F) qRT-PCR quantification of *Mad1l1* mRNA expression after 1 week of 6 mg/kg dox feed. * = p<0.05. Biorender was used to generate illustrations in 2B.

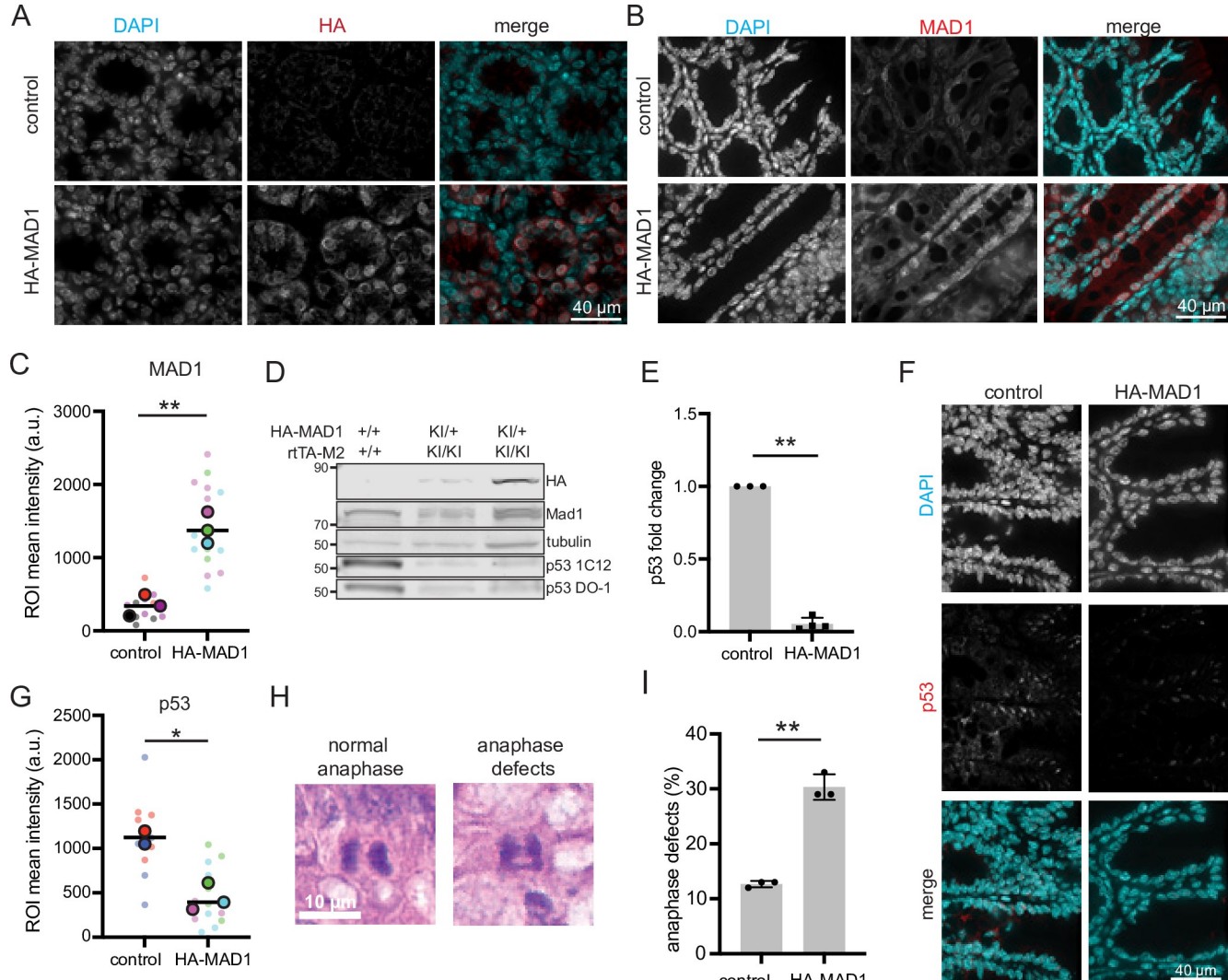

**Fig 3. Dox-inducible expression of HA-MAD1 in colon is sufficient to decrease p53 expression and cause chromosome missegregation.** (A-B) Immunofluorescence of colonic epithelium showing expression of HA (A) and increased expression of MAD1 (B) in HA-MAD1 animals after 1 week of 6 mg/kg dox feed. (C) Quantification of MAD1 expression as in B. Colors indicate samples from different animals. Large circles show mean of cells from 5 fields of view from a single animal. (D) Immunoblot showing p53 protein expression is decreased in colon tissue from 5-month old HA-MAD1 mice after 1 week of 6 mg/kg dox feed as compared to control. Lanes 2 and 3 are from different mice of the same genotype. (E) Quantification (mean +/- SEM) of p53 protein levels, based on immunoblotting with the 1C12 antibody. (F) Immunofluorescence showing p53 is decreased in the colonic epithelium after HA-MAD1 expression. (G) Quantification of p53 expression as in F. Colors indicate individual animals. Large circles show mean from 5 fields of view from a single animal. (H-I) 1 week of 6 mg/kg dox feed is sufficient to induce mitotic defects indicative of CIN in colons of 10–27 week old HA-MAD1 mice. (H) Representative H&E images of normal anaphase and anaphase with evidence of chromosome missegregation in mouse colon. (I). Quantitation (mean +/-SEM) of anaphase defects as in H. n≥20 anaphase cells per mouse in each of 3 mice/genotype. * = p<0.05. ** = p<0.001.

## Modest upregulation of MAD1 sensitizes immune-competent mice to colon lesion formation in the context of inflammation

To test whether modest MAD1 upregulation initiates colon lesion formation, we used dextran sulfate sodium (DSS) to induce inflammation in the colon. DSS is commonly used to model colitis [30,31] and colon cancer when used in combination with mutagenic agents or predisposing genetic mutations [32]. We included p53+/- animals in our experiment as a positive control, since they are predisposed to develop tumors after DSS treatment [33,34].

Experimental cohorts of control, HA-MAD1, and p53+/- animals received 2% DSS in drinking water for 4 days followed by a rest period of water for 17 days (Fig 4A). This treatment regimen was considered one cycle and was repeated for a total of 3 cycles of DSS treatment. Control cohorts in each of the 3 genotypes were given water for the entire length of the experiment. All mice in the experiment received dox feed (Fig 4A). 2% DSS treatment was sufficient to cause inflammatory damage with modest effects on weight gain (Fig C in S1 Text), as 2 cycles of 2% DSS induced morphological changes resembling that of chronic colitis compared to the organized and consistent crypt architecture observed in water-treated control colons (Fig 4B).

Serial colonoscopies were performed on the first cohorts of animals to monitor for tumor development (Fig 4C). Tumor size was assessed based on the percentage of the lumen occluded by the lesion (Fig 4D). Subsequent cohorts received colonoscopies at month 10 after starting DSS treatment (Fig 4E). 5/16 (31%) of HA-MAD1 mice and 1/7 (14%) of p53+/- DSS-treated mice developed tumors (Fig 4F). No tumors formed in control mice treated with DSS or in any of the water treated controls (Fig 4F). Tumors in HA-MAD1 mice varied in size from occluding 7–34% of the lumen of the colon (Fig 4G). 4/7 (57%) male HA-MAD1 mice but only 1/9 (11%) of female HA-MAD1 animals developed tumors (Fig 4H). These data demonstrate that modest overexpression of MAD1 is sufficient to sensitize immune-competent animals to colon tumorigenesis in the context of inflammation.

## Pathology of colon tumors

Lesions identified by colonoscopy were subjected to histological analysis by a board-certified pathologist. 2 hyperplastic polyps, 2 tubular adenomas with high grade dysplasia, and 1 mucinous adenocarcinoma were identified in HA-MAD1 mice (Fig 5 and Table 1). Two lesions were identified by colonoscopy in p53+/- mice; histological analysis revealed that one was a tubular adenoma whereas the other was a lymphoid aggregate (Table 1). Overall, DSS-treated HA-MAD1 colons showed a range of morphologies, from hyperplastic polyps to overt adenocarcinoma.

Tumors were stained for Ki-67 to assess proliferation (Fig 5). In sections without lesions, Ki-67 staining is isolated to the base of the crypts where the stem cells are located (Fig 5, 3rd column, 1st row). The extent of Ki-67 staining correlated with the morphology of the lesion and potential risk for progression to carcinoma, with tubular adenoma samples with high grade dysplasia showing positive Ki-67 staining throughout (Fig 5, 3rd column, 4th row). The mucinous adenocarcinoma showed diffuse Ki-67 expression in the tumor epithelium lining the mucin pools.

β-catenin is typically present at adherens junctions and in the cytoplasm in normal colon (Fig 5, 4th column, 1st row), but translocates into the nucleus in response to activation of the Wnt signaling pathway, which is the most common initiating event in colorectal cancer [35]. Nuclear localization of β-catenin was observed in the tubular adenomas with high grade dysplasia (Fig 5, 4th column, 4th row), but not in the mucinous adenocarcinoma (Fig 5, 4th column, 5th row). Mucinous adenocarcinomas account for approximately 10% of colorectal cancers and are less likely to have aberrant Wnt signaling and β-catenin localization [36].

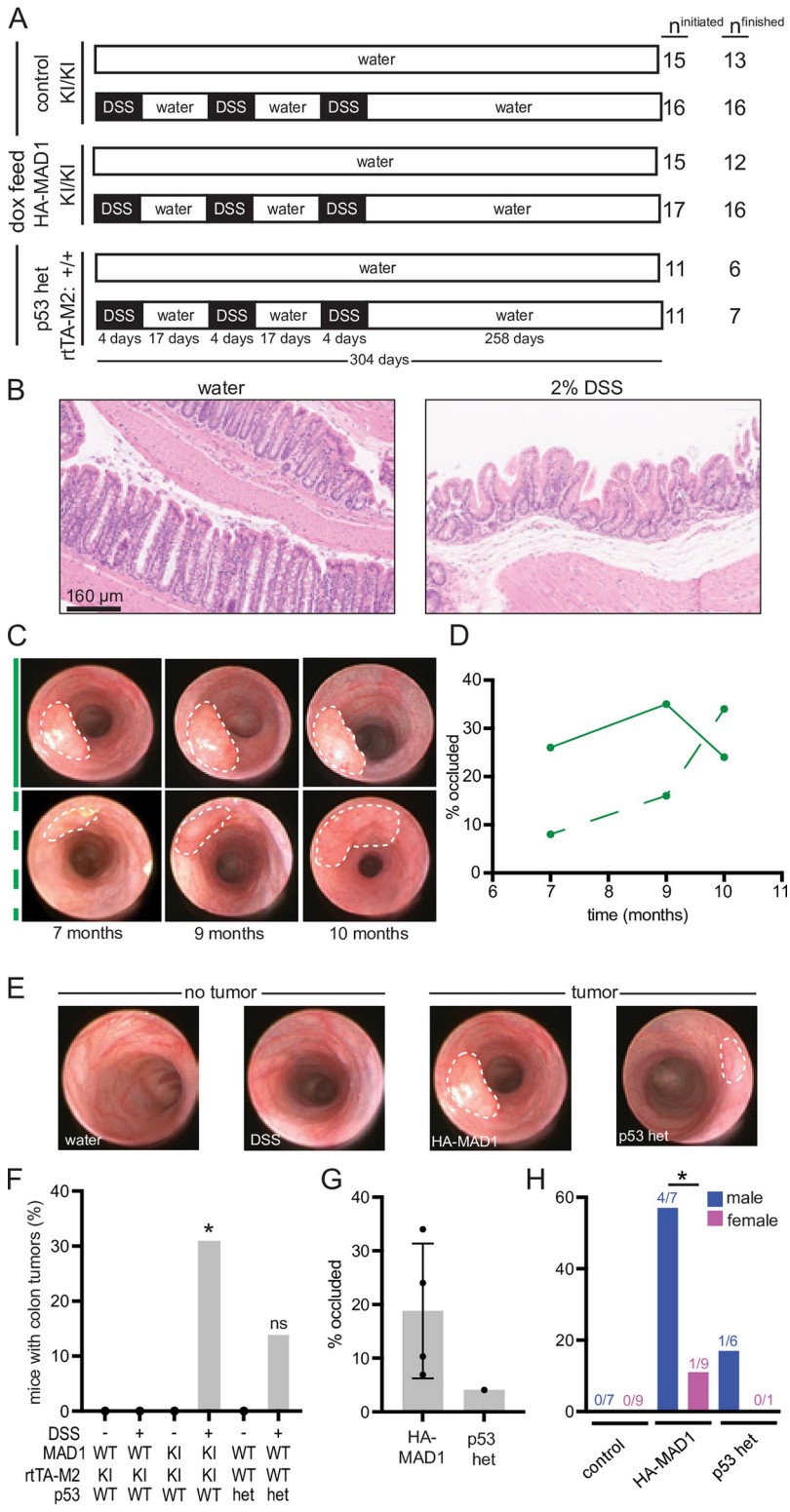

**Fig 4. Modest MAD1 upregulation sensitizes to colon tumorigenesis.** (A) Experimental schematic for DSS treatment. All mice were given dox feed for the duration of the experiment. Control animals were on water for 304 days while DSS treated animals were given 3 cycles of DSS for 4 days followed by 17 days of recovery on normal water. $n^{finished}$ indicates the number of animals that survived to 10 months after the first round of DSS in their cohort. (B) Representative H&E images of mouse colon 7 days after the second treatment with water (left) or 2% DSS (right)

showing chronic architectural changes caused by DSS treatment. (C) Representative stills from serial colonoscopies following two HA-MAD1 tumors over time. White dotted lines indicate tumor boundaries. Solid and dashed green lines indicate tumors in D. (D) Percent of colon lumen occluded by tumor over time for tumors shown in C. Solid line corresponds to the tumor in the top row of C. Dashed line corresponds to the tumor in the bottom row of C. (E) Representative stills from colonoscopy movies showing normal colon (water), scarring of the colon lining 6 months after initiation of DSS treatment (DSS), or tumors that formed 7 months after DSS treatment in HA-MAD1 and p53$^{+/-}$ animals. White dotted lines outline tumors. (F) Incidence of tumor formation detected by colonoscopy 10 months after start of DSS and confirmed histologically. Significance shown is compared to water treated control animals (*Mad1l1*$^{+/+}$;rtTA-M2$^{KI/KI}$). (G) Size of tumors at 10 months following the start of DSS observed via colonoscopy and measured as percent of colon occluded by the tumor. n = 4 tumors in HA-MAD1 animals and 1 tumor in p53$^{+/-}$ mice. (H) Colon tumor incidence after DSS treatment categorized by sex. Numbers indicate number of mice with tumor/ number of DSS-treated mice of the indicated genotype. * = p<0.05. ns = not significant.

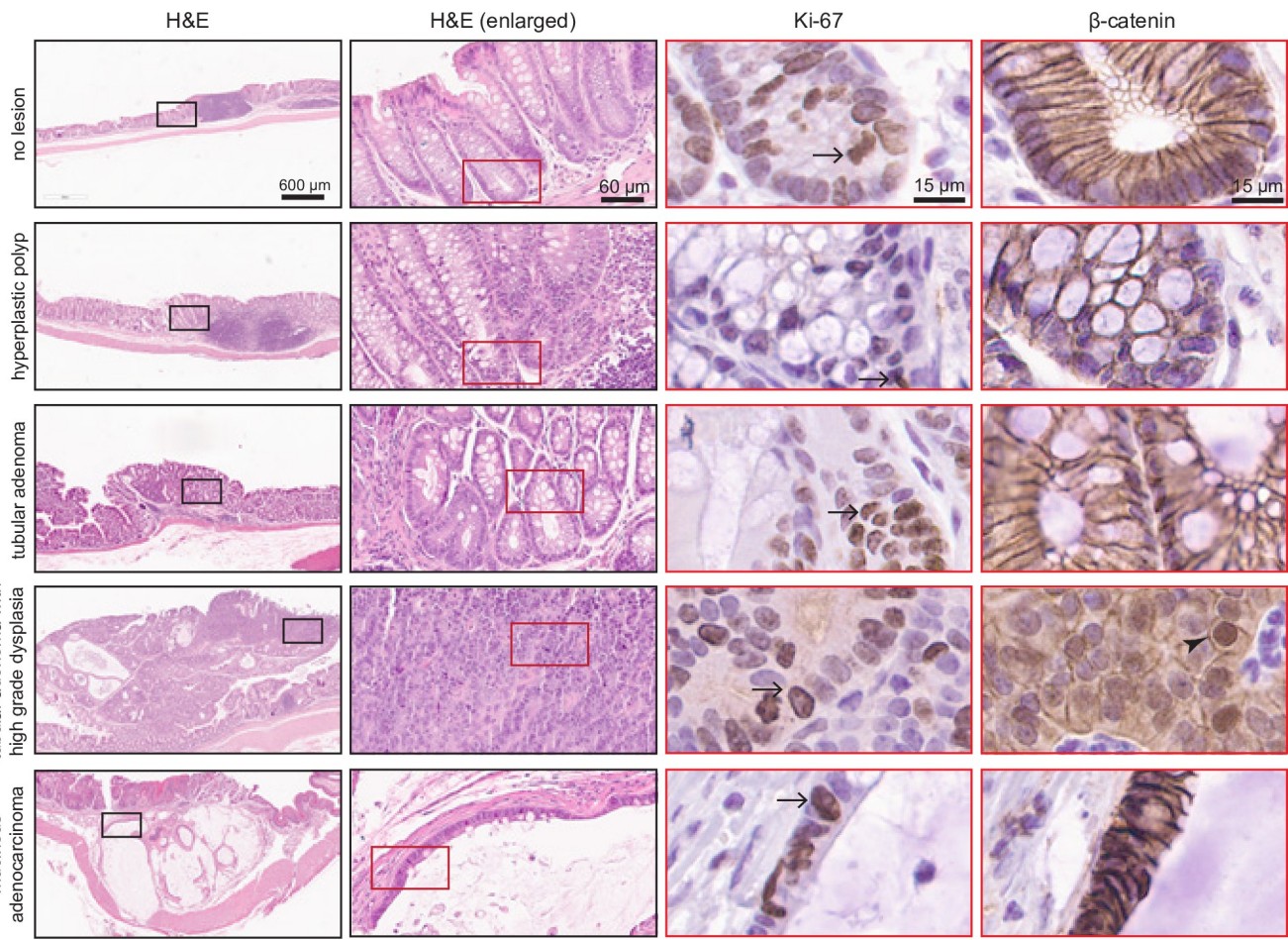

**Fig 5. Pathology of colon tumors in DSS-treated HA-MAD1 mice.** Lesions in HA-MAD1 and p53+/- colons were excised, embedded in paraffin, sectioned, and stained with H&E, Ki-67, and β-catenin. Example images of normal colon, hyperplastic polyp, tubular adenoma, tubular adenoma with high grade dysplasia, and a mucinous adenocarcinoma are shown. Black insets in the 1$^{st}$ column are enlarged in the 2$^{nd}$ column. Red insets in the 2$^{nd}$ column are enlarged in the Ki-67 and β-catenin columns. Serial sections were used for staining with H&E, Ki67 and β-catenin. Ki67 staining is confined to the base of the crypts in the no lesion control and the hyperplastic polyp. Ki67 positive cells are identified higher up in the crypts in the tubular adenoma. The tubular adenoma with high grade dysplasia loses crypt structure, but has positive Ki67 staining throughout. The adenocarcinoma had pockets of epithelial cells with positive Ki67 lining the large mucin pools. Arrows indicate cells with positive Ki-67 staininig. β-catenin staining was primarily found to be membranous and excluded from the nucleus in all samples with the exception of the tubular adenoma with high grade dysplasia, which contained β-catenin positive nuclei (arrowhead).

**Table 1. Pathologically confirmed colon lesions.**

| Genotype | no lesion | hyperplastic polyp | tubular adenoma (TA) | TA with high grade dysplasia | mucinous adenocarcinoma |
|---|---|---|---|---|---|
| control (n = 16) | 16 | 0 | 0 | 0 | 0 |
| p53+/- (n = 7) | 6 | 0 | 1 | 0 | 0 |
| HA-MAD1 (n = 16) | 11 | 2 | 0 | 2 | 1 |

### Additional tumors in HA-MAD1 mice

In addition to the distal colon, where DSS is expected to induce inflammation, we observed tumors in the small intestine and prostate of HA-MAD1 mice (Table B in S1 Text). Pathological analysis identified the small intestine tumor as a tubular adenoma (Fig D panel A in S1 Text). HA staining confirmed expression of HA-MAD1 in this tumor (Fig D panel B in S1 Text). Note that HA-MAD1 expression was also detectable in the small intestine of a 26-week old mouse after 1 week of dox feed (Fig D panel C in S1 Text). Both small intestine and prostate tumors were unexpected since the inflammatory effects of DSS are thought to be restrained to the distal colon [37,38]. Indeed, on colonoscopy we observed decreasing damage from DSS when moving proximally in the colon. Despite the lack of expected inflammatory effects outside the colon, we note that both the small intestine and prostate tumors in HA-MAD1 mice developed in DSS-treated animals. No tumors were observed in water or DSS-treated control animals (Table B in S1 Text). One DSS-treated p53+/- animal developed a splenic tumor and one water-treated p53+/- mouse developed a prostate tumor (Table B in S1 Text). These data suggest that modest upregulation of MAD1 may be sufficient for tumor promotion independent of localized inflammation in the tissue.

### HA-MAD1 tumors and tissue display mitotic defects and decreased p53 expression

Colon tumors in HA-MAD1 mice continued to show modest overexpression of MAD1 relative to control animals after 10 months of dox feed (Fig 6A and 6B). DSS treatment had minimal impact on mitotic defects consistent with CIN in control, p53+/-, or HA-MAD1 colon (Fig 6C and 6D). Interestingly, HA-MAD1 tumors showed similar rates of mitotic defects as observed in normal colon from age-matched 12-month-old HA-MAD1 mice. All HA-MAD1 colon tissue showed significantly higher rates of abnormal mitotic figures than age-matched control colon (~60% versus ~10%, Fig 6C and 6D). The rate of mitotic defects in 12-month-old HA-MAD1 mice was also substantially higher than the ~30% of cells with mitotic defects in younger HA-MAD1 mice (Fig 3I), consistent with the previously reported increase in CIN that occurs with age [39,40].

p53 expression was reduced by approximately half in HA-MAD1 tumors and adjacent normal tissue as compared to water-treated controls, similar to the level of expression in p53+/- animals (Fig 6E and 6F). This level of reduction was also similar to the level in younger HA-MAD1 animals (Fig 3G). Together, these data demonstrate that subtly increased expression of MAD1 increases mitotic defects consistent with CIN, decreases p53 expression, and sensitizes to tumorigenesis in animals with an intact immune system.

### Discussion

In this study, we developed a novel GEMM to test whether modest overexpression of MAD1, which is commonly observed in colon cancer, is sufficient for colon tumorigenesis. As predicted from studies in cellular models, we show that mice with modest (~2-fold)

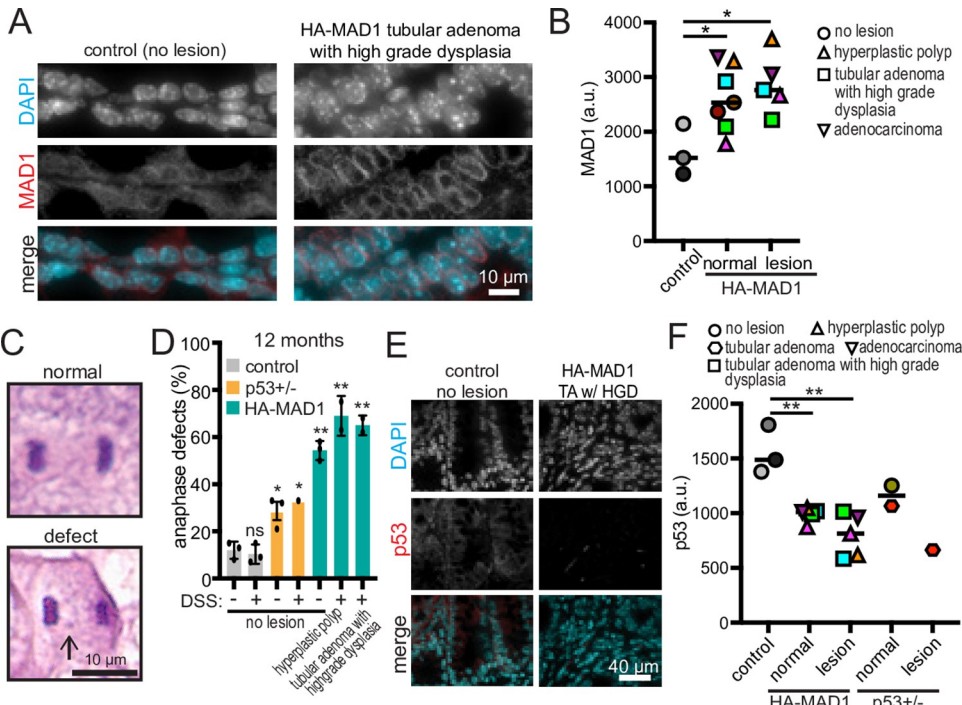

**Fig 6. HA-MAD1 tumors have increased CIN and decreased p53.** (A) Immunofluorescence showing MAD1 expression in colonic epithelium. (B) Quantification of MAD1 expression as in A. Each color represents 1 animal. Shapes indicate type of lesion. Each dot represents the average of 5 fields of view. (C) Representative H&E images of normal anaphase and defective anaphase consistent with CIN in mouse colon isolated from 12-month-old mice 10 months after the start of DSS treatment. (D). Quantitation (mean +/- SEM) of anaphase defects as in C. n≥20 anaphase cells per mouse in each of 3 (no lesion) or 2 (hyperplastic polyp, tubular adenoma with high grade dysplasia) mice per genotype. Statistics shown are in comparison to water treated control mice. (E) Immunofluorescence showing p53 expression in colon epithelium. TA w/ HGD = tubular adenoma with high grade dysplasia. (F) Quantification of p53 expression as in E. Each color represents 1 animal. Shapes indicate type of lesion. Each dot represents the average of 5 fields of view. "Normal" indicates normal tissue adjacent to the lesion. * = p<0.05. ** = p<0.01.* ns = not significant.

overexpression of MAD1 have an increase in mitotic defects that contribute to CIN and decreased p53 protein expression. Moreover, modest MAD1 upregulation sensitized immune-competent mice to colon tumor formation in the context of inflammation.

In humans, inflammation is a risk factor associated with a higher incidence of colon cancer in males than females [41]. Consistent with this, colon tumor formation was particularly pronounced in male HA-MAD1 mice. While 4/7 (57%) of male HA-MAD1 mice formed colon tumors, only 1/9 (11%) of female animals did. DSS treatment has previously been noted to cause more substantial damage, such as decreased colon length, greater weight loss, and more severe changes in stool consistency, in male animals [38,42]. In our cohort, DSS reduced weight gain specifically in male mice, though this difference was driven largely by the final timepoint. Previous studies have demonstrated that estrogen is protective against DSS-induced damage, and 17-β-estradiol treatment rescued phenotypes in male mice back to control levels [42]. Thus, the increased incidence of tumors in male mice in our model recapitulates the sex specific difference observed in human.

MAD1 upregulation increased mitotic defects consistent with CIN and decreased p53 expression, which are both associated with tumor promotion. Increased expression of MAD1 has previously been shown to induce CIN by weakening the mitotic checkpoint [3,21]. Excess MAD1 binds MAD2 in the cytoplasm, thereby reducing the pool of free MAD2 that is

converted into active inhibitors of the APC/C. MAD1 upregulation destabilizes p53 by displacing MDM2 from PML, allowing MDM2 to ubiquitinate p53 [9]. HA-MAD1 mice had increased tumor incidence relative to p53+/- animals, though this difference did not reach statistical significance, suggesting that MAD1 overexpression may confer tumor promoting activity beyond destabilization of p53. This activity could be due to the increase in mitotic defects, which have been implicated in tumor promotion [5,6]. However, the affect of CIN on tumors is complex, and CIN can also be tumor suppressive, either through induction of cell death [43–45] or activation of the immune system [7,8]. α5 integrin expression is commonly upregulated in primary tumors, and binding of α5β1 integrin to its ligand fibronectin promotes transformation and tumor cell survival [46,47]. Thus, additional tumor promoting activity from MAD1 upregulation could occur due to increased secretion of α5 integrin [48,49], or from as yet undiscovered roles of MAD1. One important area of future research will be in parsing the relative contributions of the specific functions of MAD1 responsible for tumor promotion.

It is interesting to note that we also observed tumors developing in the small intestine and prostate of HA-MAD1 mice. DSS has not been reported to induce inflammation in these tissues [38,42], suggesting these tumors are independent of inflammation. Small intestine and prostate may be particularly susceptible to the tumor promoting activities of MAD1 upregulation. However, we cannot exclude the possibility that DSS treatment caused a low level of inflammation in these tissues, and tumors outside the colon only arose in HA-MAD1 animals treated with DSS. Future studies are therefore necessary to determine if fully spontaneous tumors form in these mice in the absence of exogenously induced inflammation.

Overall, this work reports a novel GEMM with inducible, modest overexpression of MAD1. MAD1 upregulation confers tumor-promoting phenotypes and is sufficient for colon tumor formation in the context of inflammation in the presence of an intact immune system. These data implicate MAD1 as a novel prognostic marker and potential therapeutic target in colon cancer.

## Materials and methods

### Ethics statement

This study was conducted under approved protocols by the Institutional Animal Care and Use Committee of University of Wisconsin-Madison in compliance with policies established by the Office of Laboratory Animal Welfare at the National Institutes of Health (Protocols: M005342, M005601). All animal studies were performed in compliance with all relevant ethical regulations for animal testing and research. All animals were housed in a specific pathogen-free facility with water and standard chow diet *ad libitum*, unless noted. Mouse stocks were maintained on Teklad Global 2020X Rodent Diet (Inotiv, Lafayette, IN) prior to doxycycline feed, as described in the "Dextran Sodium Sulfate (DSS) Treatment" methods section.

### Generation of *Mad1l1* GEMM

Gene edited mice were generated by the University of Wisconsin Genome Editing and Animal Models (GEAM) core facility. One-cell C57BL/6J fertilized embryos were microinjected with a mixture of 25 ng/μL each of sgRNA (2 total), 40 ng/μL Cas9 protein (PNA Bio, Newbury Park, California) and 216 ng/uL donor plasmid. sgRNA sequences were: 5'-CCTGATCGTTTACT CCATAA-3' and 5'-CCATCACAAGGATCCCTATA-3'. Injected embryos were transplanted into pseudopregnant B6D2 F1 females. Pups were sequenced at weaning to screen for editing. HA-MAD1 founders were crossed with C57BL/6J mates and offspring sequenced to test for germline transmission. Sequence-confirmed HA-MAD1 F1 mice were crossed with strain B6. Cg Rosa26-CAG-rtTA3 knock-in (Jackson Laboratory strain #029627) and strain B6.Cg

R26-rtTA-M2 knock-in (Jackson Laboratory strain #006965) mice. Heterozygous HA-MAD1 animals were maintained by backcrossing with C57BL/6J mice.

## Genotyping

Ear punch DNA was isolated using QuickExtract DNA Extraction Solution (LGC Biosearch Technologies) according to the manufacturer's protocol. The following primers were used for genotyping TetO-HA-MAD1 knock-in animals: mMAD1-GT-KI-201bp-F, 5'–GATGTTCCA GATTACGCTTACCCATAC– 3'; mMAD1-WT-285bp-F, 5'–GGCAGTATTGGCTTCATGT-TAATGTTG– 3'; mMAD1-WT-285bp-R, 5'–CATGTGATACTCATACTGCTTCTGTAGG– 3'. All 3 primers were included in a single reaction with GoTaq DNA Polymerase (Promega) according to the manufacturer's protocol with an annealing temperature of 55°C for 35 cycles. Rosa26-CAG-rtTA3 knock-in mice were genotyped according to The Jackson Laboratory pro-tocol #20245. R26-rtTA-M2 knock-in mice were genotyped according to The Jackson Labora-tory protocol #29915. B6.129S2-Trp53tm1Tyj/J (Jax #002101) were genotyped according to The Jackson Laboratory protocol #27521.

## qRT-PCR

Tissue (50–100 mg) was lysed with the BeadBug microtube homogenizer (Benchmark Scien-tific). RNA was isolated using TRIzol reagent (Invitrogen) according to the manufacturer's protocol. RNA was reverse transcribed with iScript (Bio-Rad) according to the manufacturer's protocol and the cDNA was then used for real-time PCR analysis using IQ SYBR Green Super-mix (Bio-Rad) with an annealing temperature of 60°C. Primers spanning exon-exon junctions were used to detect spliced mRNA. Expression levels were calculated using the relative quanti-fication method ($\Delta\Delta$Ct). Each sample was run in triplicate and GAPDH was used as a house-keeping gene. Primers are as follows: mMad1 Forward, 5'-AGCTTGGGGTTCAGAAGC-3'; mMad1 Reverse, 5'-TCACCCTCCAGCTCCTCC-3'; mGAPDH Forward, 5'-CTCCCACTCT TCCACCTTCG-3'; mGAPDH Reverse, 5'-GCCTCTCTTGCTCAGTGTCC-3'.

## Immunoblotting

Tissue (20–50 mg) was lysed in 500 mL RIPA buffer (150 mM NaCl, 25 mM Tris, 1% NP40, 1% deoxycholic acid sodium salt, pH 7.0–7.4) supplemented with protease inhibitor (cOmplete Protease Inhibitor Cocktail, Roche) using the BeadBug microtube homogenizer (Benchmark Scientific). Proteins were separated by 12% SDS-PAGE, transferred to nitrocellulose mem-brane, blocked with 5% milk in TBST for 1 hour at room temperature. Primary antibodies were diluted in 2% BSA + 0.02% sodium azide in PBS and incubated overnight at 4°C. Second-ary antibodies were diluted in 5% milk in TBST and incubated for 45 minutes at room temper-ature. Membranes were imaged on the LI-COR Odyssey imager and analyzed using LI-COR Image Studio. Primary antibody dilutions: MAD1 (affinity purified rabbit anti-MAD1 against amino acids 333–617 [3]), 1:3000; HA (16B12, BioLegend), 1:1000; alpha-tubulin (12G10, Developmental Studies Hybridoma Bank) 1:5000. Secondary antibodies were diluted 1:10,000 (IRDye, LI-COR).

## Tissue Immunostaining

p53 and MAD1: Paraffin-embedded sections of 5 μm thickness were dewaxed by incubation in xylene (3 x 5 minutes) and rehydrated by ethanol series: 100% (2 x 2 minutes), 95% (2 min-utes), 80% (2 minutes), 70% (2 minutes), 50% (2 minutes), and water (5 minutes). Antigen retrieval was performed using Decloaking Chamber NxGen (BioCare Medical) for 15 minutes

at 110°C in citrate buffer pH 6.0 (10 mM citric acid, 0.05% Tween-20) for MAD1 and p53 antibodies and in Tris-EDTA buffer pH 9.0 (10 mM Tris, 1 mM EDTA, 0.05% Tween-20) for the HA antibody. Samples were allowed to cool to room temperature for 30 minutes, rinsed in PBS, and blocked in 10% goat serum in PBS for 1 hour at room temperature. Primary antibodies were diluted in PBS with 1% goat serum and 0.01% Triton X-100 and incubated on tissues overnight at 4°C. Tissues were washed in PBS (3 x 5 minutes) followed by incubation with secondary antibody diluted in PBS for 30 minutes at room temperature. Tissues were washed in PBS (3 x 5 minutes) and then mounted using vectashield antifade mounting medium with DAPI (Vector Laboratories, H-1200). Primary antibody dilutions: MAD1 (affinity purified rabbit anti-MAD1 against amino acids 333–617 [3]), 1:1600; p53 (ab131442, Abcam), 1:100; HA (C29F4, Cell Signaling), 1:400. Secondary antibodies were diluted 1:500 (Goat anti-Rabbit Alexa Fluor 594, ThermoFisher A-11037). Z-stack images were acquired using a Nikon Ti-E inverted microscope with a CoolSNAP HQ2 camera using a 60x 1.4 NA objective with 0.3 μm steps. Nikon Elements software was used to process and analyze the images. Max intensity projections are shown. Sections for quantification were stained contemporaneously. Quantification was performed on images collected comtemporaneously using identical exposure times. Mean fluorescent intensity of MAD1 or p53 was quantified in nuclei of 5 cells in 5 fields per sample (25 cells/sample) using DAPI to create nuclear ROIs.

Ki67 and β-catenin: Slides were heated for 5 minutes at 65°C, deparaffinized in xylenes for (2) 10 minute incubations and rehydrated using a series of ethanol dilutions (100%, 95%, 80%, 70%, 50%). Washes, blocks and primary antibody dilutions were performed with 1x PBS + 0.5% Tween 20 (1x PBST) for Ki67 stains or 1x Tris Buffered Saline + 0.5% Tween 20 (1x TBST) for β-catenin. Antigen retrieval was performed by boiling samples for 30 minutes in citrate buffer (pH 6.0) for Ki67 or 30 minutes in Tris-EDTA buffer (pH 9.0) for β-catenin. Peroxidase activity was blocked for 5 minutes at room temperature with BLOXALL (SP-6000-100, VectorLab). 5% normal goat serum (S1000-20, Vector Labs) in 5% skim milk in the appropriate salt buffer listed above for each antibody was used to block for 1 hour at room temperature and to dilute primary antibodies for anti-β-catenin (1:200, CloneD10A8, #8480S, Cell Signaling Technology) and anti-Ki67 (1:400, Clone D3B5, #12202T, Cell Signaling Technology). Primary antibodies were incubated overnight at 4°C in a humid chamber. Anti-rabbit HRP secondary antibody (RHRP520L, BioCare) was incubated for 30 minutes at room temperature. Slides were developed with diaminobenzidine (DAB; #11725S Cell Signaling Technology) prepared per the manufacturer's protocol and incubated for 45 seconds for Ki67 and 4 minutes for β-catenin. Counterstaining was conducted with CAT hematoxylin (CATHE-M, BioCare) for 30 seconds. Slides were dehydrated in the ethanol series previously described, incubated in xylenes for 10 minutes and cover slipped with permount. Slides were imaged using the Aperio AT2 digital RBG slide scanner at 40x.

## Dextran sulfate sodium (DSS) treatment

Mice between 8–11 weeks of age started on dox feed approximately one week before starting DSS or water treatment (Rodent Diet 2018, 6000 mg/kg Doxycycline; TD.01533) to ensure MAD1 overexpression. Mice were then randomly assigned to either DSS or water treatment. DSS (MW ca 40,000; Thermo Scientific Chemicals catalog number AAJ6360622, lot# Y08H016) was used to induce inflammation in colon. Each DSS cycle consisted of *ad libitum* 2% DSS dissolved in water for 4 days, followed by 17 days of regular water. After this first cycle, two additional cycles of DSS treatment were given for a total of three DSS cycles. Mice in the water group only received water and were handled on the same days as the DSS treatment mice. All mice in both treatment groups were weighed at the start and end of each 4 day period

to monitor for weight loss and examined at least once daily to identify signs of distress (i.e. hunched posture, bloating, pallor, etc.). All mice were euthanized when moribund or 10 months after the start of the first treatment cycle.

## Colonoscopy tumor surveillance

The first two cohorts of mice received serial colonoscopies at 7, 9, and 10 months after beginning DSS to observe tumor formation in the distal colon and to determine the appropriate endpoint. All other cohorts received colonoscopies at the 10 month time point. The Karl Storz Coloview system was used as previously described [50]. In brief, mice were weighed to assess for weight loss, anesthetized with 2% isoflurane and the distal colon was gently cleared of fecal pellets by 1x PBS enemas. A compressed air pump on the operating sheath insufflated the colon, and the colonoscope was inserted approximately 4 cm into the distal colon to carefully examine the epithelial lining with bright light. Digital video and still images of the mucosa were recorded as the colonoscope was slowly withdrawn. Mice were monitored post-procedurally and allowed to recover from anesthesia without complication. Still images of tumors were analyzed for the percentage of colon occluded by the tumor using Image J.

## Histopathological analysis

Pathological diagnosis of each colon tumor was determined by a board-certified pathologist (K.A.M.) blinded to treatment regimens using an Olympus BX43 microscope. Slides were imaged using the Aperio AT2 digital RBG slide scanner at 40x.

## Statistical analysis

Statistal analysis was performed using GraphPad Prism (Version 10.2.3). Significance was assed using an unpaired Student's *t* test, unless otherwise noted. Superplot statistical analysis was run on means (large dots) rather than individual values (small dots) in Fig 3C and 3G. Two-tailed chi-square test was used for Fig 4F and 4H, and Table A in S1 Text. Sen-Adichie test for parallelism was used in Fig C in S1 Text and was performed using MSTAT (https://oncology.wisc.edu/mstat/).

## Supporting information

**S1 Text. Supporting Figures and Tables. Fig A. Generation and validation of dox-inducible HA-MAD1 mice.** (A) Schematic showing targeting strategy for editing MAD1 (encoded by the *Mad1l1* gene, left) and edited allele (right). Two gRNAs were used to cut on either side of exon 2, the first coding exon of mouse *Mad1l1*. The edited allele of *Mad1l1* includes a TRE3G promoter and an HA tag after the first ATG, which is in exon 2. (B) Schematic of method used to generate dox inducible HA-MAD1 mice. Cas9, 2 sgRNAs, and donor DNA were injected into C57BL/6 zygotes, which were then transferred to a pseudopregnant mouse. Three founder mice had the intended edit, though 1 founder had an additional 13 base-pair deletion in the intronic region after exon 2. Founder mice were bred with a C56BL/6 mouse and the resulting F1 mice were genotyped to ensure germline transmission of the edit. The schematic was generated in Biorender. (C) Genotyping strategy with three primers used to screen for edited mice. (D) Agarose gel showing successful editing in founder mice. KI = Knock In. The founder with a 13-base pair indel is indicated by KI*. NC = No DNA control. (E) Agarose gel showing germline transmission of the edited alleles in the F1 generation. **Fig B. CAG-rtTA3 results in dox-inducible expression of HA-MAD1 RNA.** (A) Breeding strategy to produce mice that are doubly heterozygous for HA-MAD1 and CAG-rtTA3 (Jax strain #029627). Doubly

heterozygous HA-MAD1$^{KI/+}$;CAG-rtTA3$^{KI/+}$ animals were used in B-G. Doubly heterozygous mice were bred with animals heterozygous for CAG-rtTA3 to produce HA-MAD1$^{KI/+}$;CAG-rtTA3$^{KI/KI}$ animals in E-F. Biorender was used to generate the schematic. (B) qRT-PCR of *Mad1l1* mRNA showing substantial increase in *Mad1l1* mRNA expression (spliced transcripts) in spleen and fat after 1 week of 625 mg/kg dox feed. (C-D) MAD1 protein expression in mice after 1 week on 625 mg/kg dox feed. Red arrows in immunoblot (C) indicate HA-MAD1. (D) Quantification of MAD1 protein level as assessed by immunoblot. (E-F) Immunoblot (E) and quantitation (F) showing that mice homozygous for CAG-rtTA3 do not exhibit increased expression of dox-inducible HA-MAD1 relative to mice heterozygous for CAG-rtTA3. (G) Use of dox feed for 1 week and 1 month result in similar levels of MAD1 expression in spleen tissues isolated from HA-MAD1$^{KI/+}$;rtTA-M2$^{KI/+}$ mice. (H) Immunoblot showing expression of HA and MAD1 in control or HA-MAD1 colon with or without dox feed. HA-MAD1 is not expressed in the absence of dox feed. **Fig C. Effects of DSS treatment on weight.** Mice weight as a percentage of starting weight on day 0 +/- SEM. DSS treatment occurred on days 0–4, 21–25, and 42–46. Mice were weighed at the beginning and end of each cycle. (A) Female mice have similar weight gain in water or DSS treatment groups. (B) DSS treatment impaired weight gain in male mice. (C-D) Data from A-B separated by sex rather than treatment group. ns = not significant; $^*$ = p $<$ 0.05. **Fig D. HA-MAD1 expression in small intestine.** (A) H&E images of normal small intestine and tubular adenoma in HA-MAD1 mouse. Normal indicates normal adjacent tissue. (B) Immunofluorescence showing expression of HA-MAD1 in small intestine tubular adenoma. (C) Dox-inducible expression of HA-MAD1 in small intestine. Tissue for immunoblot was collected after 1 week of dox feed (without DSS treatment). MAD1 upper band (red arrow) corresponds with HA band. **Table A.** Homozygous HA-MAD1 mice are embryonic lethal in the absence of dox. **Table B.** Potentially inflammation-independent tumors 10 months after cohorts initiated DSS treatment. (PDF)

**S1 Data. Excel file containing source data for Figs 1–4, 6, B and C in S1 Text.** (XLSX)

## Acknowledgments

We thank Kathy Krentz and Dr. C. Dustin Rubinstein in the UW Genome Editing and Animal Models Core Facility (GEAM) for generating HA-MAD1 mice, Conrad Blosch in the Biomedical Research Model Services (BRMS) Rodent Breeding Core and Research Services Core for assistance with mouse colony management, Magdaline Baus for her assistance with DSS treatments, Toshi Kinoshita in the Translational Research Initiatives in Pathology (TRIP) lab for histology services, and members of the Weaver, Halberg, Burkard, Suzuki, and Cosper laboratories for insightful discussions. The authors thank the UWCCC Shared Resources GEAM and University of Wisconsin Translational Research Initiatives in Pathology (TRIP), supported by in part by UWCCC (P30 CA014520), for use of their facilities and services.

## Author Contributions

**Conceptualization:** Sarah E. Copeland, Santina M. Snow, Jun Wan, Richard B. Halberg, Beth A. Weaver.

**Data curation:** Sarah E. Copeland, Santina M. Snow.

**Formal analysis:** Sarah E. Copeland.

**Funding acquisition:** Richard B. Halberg, Beth A. Weaver.

**Investigation:** Sarah E. Copeland, Santina M. Snow, Jun Wan, Kristina A. Matkowskyj.

**Methodology:** Sarah E. Copeland, Santina M. Snow, Jun Wan, Richard B. Halberg, Beth A. Weaver.

**Supervision:** Richard B. Halberg, Beth A. Weaver.

**Writing – original draft:** Sarah E. Copeland, Beth A. Weaver.

**Writing – review & editing:** Sarah E. Copeland, Santina M. Snow, Jun Wan, Kristina A. Matkowskyj, Richard B. Halberg, Beth A. Weaver.

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
