## [Decision Letter · Decision Letter 0]

22 Jul 2024

Dear Dr Weaver,

Thank you very much for submitting your Research Article entitled 'MAD1 upregulation sensitizes to inflammation-mediated tumor formation' to PLOS Genetics.

The manuscript was fully evaluated at the editorial level and by independent peer reviewers. All three reviewers found your work to be well-conceived and executed; they agree that it presents impactful findings of broad interest. Their comments indicate that the manuscript needs only minor revisions. Some edits are needed to ensure that the discussion better aligns with the data (see reviewer #3). Reviewers #1 and 2 suggests some modifications to the figures so that they are more accessible to cell biologists and cancer biologists.

We therefore ask you to modify the manuscript according to the review recommendations. Your revisions should address the specific points made by each reviewer.

To resubmit, log into your Editorial Manager account and select the option 'Revise Submission' in the 'Submissions Needing Revision' folder.

Yours sincerely,

Ajit P. Joglekar, Ph.D.

Guest Editor

PLOS Genetics

David Kwiatkowski

Section Editor

PLOS Genetics

Reviewer's Responses to Questions

**Comments to the Authors:**

Reviewer #1: In the current manuscript the authors develop a novel mouse model that has modest overexpression of the important spindle assembly checkpoint protein Mad1. Previous studies have shown that Mad1 is overexpressed in multiple tumor types and there is some evidence that it can induce chromosome mis-segregation as well as destabilize p53, two events that are pro-tumorigenic. In the present study, the authors use CRISPR genome editing to introduce a doxycycline inducible promotor and HA tag into the endogenous Mad1l1 gene locus in mice, such they were able to induce expression by feeding the mice doxycycline. This resulted in ~ 2-fold increase in expression of the Mad1p, which resulted in a decrease in p53 and an increase in chromosome segregation errors. They next treated the animals with a colon-specific inflammatory agent, which induced colon tumors in 31% of the mice, suggesting that overexpression of Mad1 can cause tumors in response to inflammation in immune-competent mice. Overall, the findings are important and will be of interest to the readership of PLOS Genetics, as Mad1 is a critical player in the spindle assembly checkpoint, and there is ample evidence that its mis-expression is correlated with tumorigenesis. This paper is largely the characterization of this mouse, which is necessary before this valuable tool can be utilized by the research community. I am in strong support for publication after the authors address the comments below.

General Comments:

1. My main concern with the paper is that the number of tumors formed is small, making it difficult to draw strong conclusions in some of the experiments or to do robust statistics. The authors started out with a reasonable sample size (16 each for control and experimental), but then only a percentage of the mice formed tumors. It would be crazy to repeat the entire study to increase sample size when it is unlikely that any of the major conclusions would change. To address this concern, there are several places where the authors should change their wording to be slightly more cautious on their interpretations- specific examples are listed below.

2. A second general comment is that the authors can improve clarity and presentation with better labeling of figures and descriptions in figure legends. Their work is at the intersection of cancer biology and cell biology. Many of the tissue images and the pathology analysis have no arrows or other indicators to draw the reader’s attention to specific areas. In addition, in many places the figure legends are written as results or conclusions rather than stating specifically what is shown in the figure. Again, specific examples will be given below, but overall, the authors should revisit all of the figures and legends with a focus on improving clarity.

Specific Comments:

1. I understand that the authors have genetically engineered the mice with the dox-inducible promotor to overexpress Mad1, but I find the comparison to mice with WT Mad1 to be a bit confusing. The overexpressed Mad1 is also WT- there is just more of it. When someone compares something to WT, my first thought is always a mutant vs. WT gene. Please consider changing the nomenclature for the WT control. Examples I can think of is Mad1 control vs. Mad1-OE or some variation of the control rtTA. These are just suggestions, and I leave it to the authors to choose the clearest option.

2. Throughout the paper the authors use the term CIN to describe the defects in chromosome segregation. Technically, they have only showed anaphase defects and have not demonstrated true CIN, which occurs through multiple cell divisions.

3. Please add the specific study number (numbers) for the TCGA data used in Figure 1.

4. There is inadequate description of how the increased fluorescence signal was measured. There is no detail in the methods, and the figure legend is insufficient.

5. Curious in Fig. 2F why the increased ratio of mRNA/protein is not maintained in the skin- perhaps it is just that there are only two skin samples so insufficient to make a strong conclusion.

6. For Figure 3B, in the merged image in the red channel still looks grey. If it is indeed red, then the images need to be changed to have increased contrast.

7. Figure 1D labels are not clear: there are two lanes of the same thing, but the blots are different. Also, for the p53 labels, I am assuming that the two blots are the different antibodies, but then which one is used for quantification in E? And for the IF in F and G- I realize that the layout doesn't fit, but the quantification in F comes from the images in G- thus the data in F needs to be after G. I understand the layout problems- but having figures out of order is confusing for the reader.

8. I disagree with the conclusion that the effects of weight are more pronounced in males. The difference comes down to a single measurement at the latest time point; otherwise, the curves are essentially the same. In addition, the difference between males and females on water (C) is the same magnitude as the DSS effect comparison in (D).

9. I appreciate that you started with a good number of mice and ended up with much fewer, but I don't think you have a sufficient number of tumors to conclude that there is significance to the finding that there were more tumors in male mice- especially when you consider that 14% of tumors in p53hets in panel F is not significant- because what it represents is one mouse. Please soften the wording to suggest a trend rather than a strong conclusion.

10. In the description of the tumors in Figure 5, it describes the pathology on 5 tumors, but the data from figure 4 say there were 7 tumors. What happened to the other ones?

11. For the data in Figure 5, you need arrows, arrowheads and other labels to illustrate this figure. This work is important to both cancer biologists, as well as to cell biologists who may have less experience in pathology. Also, the Ki67 staining is very difficult to detect in these images at the presented magnification because it is not all that high contrast. Please consider scaling to enhance the contrast and visibility of the staining.

12. I am unable to discern the differences in localization of the B-catenin staining at this magnification. Please include insets with increased magnification to better illustrate this point.

Reviewer #2: This study describes the impact of a modest (2-fold) overexpression of MAD1 in the mouse. Decreased p53 expression and increased chromosomal instability (CIN) are observed in the colon. When combined with exposure to the inflammatory agent dextran sulphate sodium (DSS), 31% of mice develop colon lesions and cancers are observed. More lesions are observed in males. In addition, lesions are observed in tissues (small intestine and prostate) not directly impacted by DSS-induced inflammation.

This is a well controlled study, carefully examining the impact of mild MAD1 overexpression in a novel genetically-engineered mouse model. The manuscript is very clearly written. Once a few issues with the figures and legends have been clarified, I will be happy to recommend publication.

Figure 3: how low are the p53 levels in the colonic epithelium? No numbers are given in the text. The datapoints in Fig3E-G look close to zero.

MAD1 levels seem to vary significantly from one mouse to another. Eg. Relatively low in the light blue mouse (Fig.3C): do MAD1 expression levels correlate with phenotypes (p53 levels and CIN)? It appears not from panels F and I. Why don’t they (in particular for the light blue mouse)?

Discussion (p10): ‘beyond stabilization of p53’. The authors should expand this section of their discussion, saying how the excess Mad1 protein might perturb chromosome segregation leading to CIN. Similarly, a little more speculation on why tumors are developing in the small intestine/prostate, but only in the mice treated with DSS, will be of general interest.

Other points, generally improving figures and legends:

Figure 1: the labelling should be improved and clearly explained in the figure legend. egs. RFS and PPS can be written in full. Q1 (lower quartile) is labelled as ‘low’ in B and C? Q4 as high? The labelling and description in the legend should be more consistent. HR is ??

Figure 2: can the Mad1 western be improved (Fig.2D)? Particularly in the skin, the Mad1 band is very weak, with several other bands nearby.

Figure 6: Panel D, color should be added to the graph so that individual data-points can be seen (the greys are too dark here, in the mutants). Too many abbreviations are being used here for the general reader. (TA/HGD/HP).

Figure S2. Western: what are the striking bands observed with anti-Mad1 in the first three fat samples?

Reviewer #3: In this study by Copeland et. al., the authors aimed to investigate the role of Mitotic Arrest Deficient 1 (MAD1L1, MAD1) up regulation, in sensitizing the colon to cancerous lesions. Prior work from this group has shown that MAD1 up regulation in cell lines leads to chromosome instability and destabilization of p53. In this study, the authors successfully generated a mouse model with tet-inducible MAD1 to study the effects of MAD1 up regulation in the colon. They show that a modest 2-fold up regulation in MAD1 leads to chromosome instability and p53 down regulation. MAD1 up regulation, when paired with DSS-induced inflammation, also led to the development of tumorous lesions in the colon, small intestine and prostate. The authors propose that MAD1 up regulation in the colon is sufficient to promote tumorigenesis in inflamed colonic epithelium.

The authors have developed and characterized an elegant tool to generate modest and inducible protein up regulation in-vivo. They have used it to then answer precise questions in a well-designed manner and the conclusions are supported by the data presented. I am overall enthusiastic about the publication of this study in PLOS Genetics. I would only suggest some modifications to the text and the figures for the purpose of making the work accessible to non-experts of the field.

Comments

1. In the section where the authors describe the generation of inducible HA-MAD1 mice, the switch between the Rosa26-CAG-rtTA3 and Rosa26-rtTA-M2 systems is very abrupt. A description of these systems is critical to understand their differences in inducing the up regulation of HA-MAD1 mRNA and protein levels.

2. All the data shown in figure 2 and the following figures are from mice fed with Dox chow. Though the fact that a homozygous TetO-HA-MAD1 could not be generated suggests that the promoter is suppressed, it would be important to show the basal expression of HA-MAD1 in the absence of Dox chow or, at the very least, explain why there are no experiments where the “no Dox” condition is present.

3. Could the authors also comment on why the p53+/- cohort has only one female mouse (figure 4H)? Also, the text states that “14% of p53+/- mice developed tumors”, even though this represents only 1 out of 7 mice and the increase is, according to the author’s analysis, not significant (figure 4F). The authors must tone this statement down.

4. In Figure 3D, could the authors please clarify what difference between lanes 2 and 3 are.

5. In Figure 4H could the authors please comment on whether the difference in incidences of lesions between male and female mice could be because of the extent of DSS induced inflammation itself? It is important to address this as there is evidence to suggest that DSS affects males more than females (Figure S3 and Chassaing B et. al., 2014).

6. The authors claim that the tumors in the small intestine and the prostate in DSS treated HA-MAD1 mice are independent of inflammation. To conclude this, the authors must either show that there are no markers of inflammation present in these tissues (low level inflammation could still be present in the absence of any widespread morphological changes) or tone this statement down.

7. In the section titled “HA-MAD1 tumors display mitotic defects and decreased p53 expression” the text and data presented indicates that there is not much difference in mitotic defects or p53 levels between tumor and normal colon tissues. The authors should title the section to better represent the data and conclusions presented.

8. Table S2, Could the authors please separate water and DSS treated animals. This would make the data easier to understand/interpret and align better with the text that references this data.

**Have all data underlying the figures and results presented in the manuscript been provided?**

Reviewer #1: Yes

Reviewer #2: Yes

Reviewer #3: Yes

PLOS authors have the option to publish the peer review history of their article (what does this mean?). If published, this will include your full peer review and any attached files.

Reviewer #1: No

Reviewer #2: No

Reviewer #3: No

---

## [Editor Report · Decision Letter 1]

23 Sep 2024

Dear Dr Weaver,

We are pleased to inform you that your manuscript entitled "MAD1 upregulation sensitizes to inflammation-mediated tumor formation" has been editorially accepted for publication in PLOS Genetics. Congratulations!

Yours sincerely,

Ajit P. Joglekar, Ph.D.

Guest Editor

PLOS Genetics

David Kwiatkowski

%CORR_ED_EDITOR_ROLE%

PLOS Genetics

Comments from the reviewers (if applicable):

**Data Deposition**

http://datadryad.org/submit?journalID=pgenetics&manu=PGENETICS-D-24-00657R1

**Press Queries**

---

## [Editor Report · Acceptance letter]

3 Oct 2024

PGENETICS-D-24-00657R1 

MAD1 upregulation sensitizes to inflammation-mediated tumor formation 

Dear Dr Weaver, 

We are pleased to inform you that your manuscript entitled "MAD1 upregulation sensitizes to inflammation-mediated tumor formation" has been formally accepted for publication in PLOS Genetics! Your manuscript is now with our production department and you will be notified of the publication date in due course.

With kind regards,

Anita Estes

PLOS Genetics

On behalf of:
